# Explainable Graph Learning for Particle Accelerator Operations

**Song Wang**  *sw3wv@virginia.edu*
*Department of Electrical and Computer Engineering*
*University of Virginia*

**Christopher Tennant**  *tennant@jlab.org*
*Thomas Jefferson National Accelerator Facility*

**Jundong Li**  *jundong@virginia.edu*
*Department of Electrical and Computer Engineering*
*University of Virginia*

**Reviewed on OpenReview:** `https://openreview.net/forum?id=jnReRk2EX1`

## Abstract

Particle accelerators are vital tools in physics, medicine, and industry, requiring precise tuning to ensure optimal beam performance. However, real-world deviations from idealized simulations make beam tuning a time-consuming and error-prone process. In this work, we propose an explanation-driven framework for actionable insight into beamline operations, focusing on the injector beamline at the Continuous Electron Beam Accelerator Facility (CEBAF). We represent beamline configurations as heterogeneous graphs, where setting nodes represent elements that human operators can actively adjust during beam tuning, and reading nodes passively provide diagnostic feedback. To identify the most influential setting nodes responsible for differences between any two beamline configurations, our approach first predicts the resulting changes in reading nodes caused by variations in settings, and then learns importance scores that capture the joint influence of multiple setting nodes. Experimental results on real-world CEBAF injector data demonstrate the framework's ability to generate interpretable insights that can assist human operators in beamline tuning and reduce operational overhead.

## 1 Introduction

Particle accelerators are complex scientific instruments designed to accelerate charged particles, such as protons and electrons, to high energies (Wille, 2000; Conte & MacKay, 2008; Sessler, 2014; Wiedemann, 2015; Appleby et al., 2020). These devices have found application in a broad range of disciplines, such as particle physics, nuclear physics, and materials science, which contribute to uncovering the fundamental properties of matter and the universe (Brüning & Myers, 2016). Accelerators can probe nuclear structure (Leemann et al., 2001; Willeke & Beebe-Wang, 2021), characterize material properties, and support medical fields through radiation therapy and advanced imaging technologies (Peach et al., 2011; Podgoršak, 2014; Flanz, 2015).

As some of the most advanced and complex machines ever engineered, particle accelerators demand precise operation (Kutsaev, 2021; Friedrich et al., 2017). High-fidelity beamline simulations are commonly used to assist human operators in managing accelerator performance. However, these simulations are idealized and often differ from the complexities of real-world implementations. This mismatch necessitates beam tuning, an iterative process that seeks to align operational parameters with simulated performance. Beam tuning is frequently time-consuming and challenging, contributing significantly to machine downtime. The current reliance on limited diagnostics and simulation interactions compounds this inefficiency.

In this work, we introduce a novel explanation-driven framework for achieving actionable insights into complex, high-dimensional systems by leveraging graph-based representations of accelerator beamline configurations. The beamline is modeled as a heterogeneous graph where nodes represent different elements, divided into setting nodes (which can be actively adjusted by human operators) and reading nodes (which are observable but not tunable). Directed edges capture the beamline's topology, where influence flows from upstream to downstream nodes. Each node is associated with a unique feature vector representing its state. However, two key challenges arise in this setting. First, only the setting nodes can be adjusted, while the reading nodes only passively reflect system outcomes. This imposes a constraint: it is infeasible to freely vary setting node features, as the resulting changes in the reading nodes are not directly known. Second, adjusting multiple setting nodes simultaneously can lead to complex, nonlinear interactions, making it difficult to disentangle their individual contributions and interpret the resulting changes in reading node features. As a result, explaining the effects of specific adjustments remains a significant challenge. Understanding these effects is essential for effective beamline tuning.

To address these challenges, our method identifies and ranks influential nodes based on their contribution to beamline changes. To isolate the impact of setting nodes, we train a representation learner that predicts reading node features based solely on settings, thus enabling a clean comparison between predicted and actual outcomes. Second, to model the joint influence of multiple setting nodes, we introduce trainable node masks with regularization that help isolate the most influential adjustments. These masks are optimized to highlight setting nodes that significantly impact downstream behavior, providing interpretable node importance scores that can guide beam tuning. Moreover, evaluating our framework is challenging due to the absence of ground truth for node importance. To address this, we estimate setting node influence by measuring how individual adjustments affect the prediction error on reading nodes. This provides a principled proxy for assessing explanation accuracy. The main contributions of this work are as follows:

1. **Novel Representation**: We model particle accelerator beamlines as heterogeneous, directed graphs, where nodes represent beamline elements and edges capture their interactions. This enables effective application of graph-based machine learning techniques for beamline analysis.

2. **Explanation Mechanism**: We propose an explanation-driven framework that identifies the most influential setting nodes responsible for observed beamline changes. This is achieved by training a node-mask mechanism with regularization based on predicted versus actual reading node changes.

3. **Practical Evaluation**: Our approach is validated on real-world beamline datasets. To achieve ground truth for evaluation, we design an efficient method to independently measure the impact of each node. The results demonstrate its ability to identify important changed nodes and provide interpretable insights.

By focusing on explanations, our framework provides a practical tool for improving particle accelerator operations while empowering operators with a greater understanding of a high-dimensional, complex system.

## 2 Preliminaries

### 2.1 Data Preparation

The Continuous Electron Beam Accelerator Facility (CEBAF), located at the Thomas Jefferson National Accelerator Facility (JLab), is a powerful recirculating linear accelerator capable of simultaneously delivering electron beams to four nuclear physics experimental stations (Reece, 2016; Adderley et al., 2024). A schematic of CEBAF is shown in Figure 1a. For this study, we focus specifically on the injector beamline. The injector serves as an excellent testbed due to its relatively compact scale, diversity of beamline components, and extensive archived tuning data. This region is critical to beam performance and serves as a focal point of operational adjustments, providing abundant data for the development of data-driven models. To show how a beamline can be represented as a graph, consider the example illustrated in Fig. 1b. The beamline comprises various element types, including beam current monitors (BCMs), beam position monitors (BPMs), quadrupoles (Q), solenoids (SOL), and correctors (CORR), all represented as nodes in the graph. Each node type is characterized by unique features: quadrupoles, correctors, and solenoids have a single value

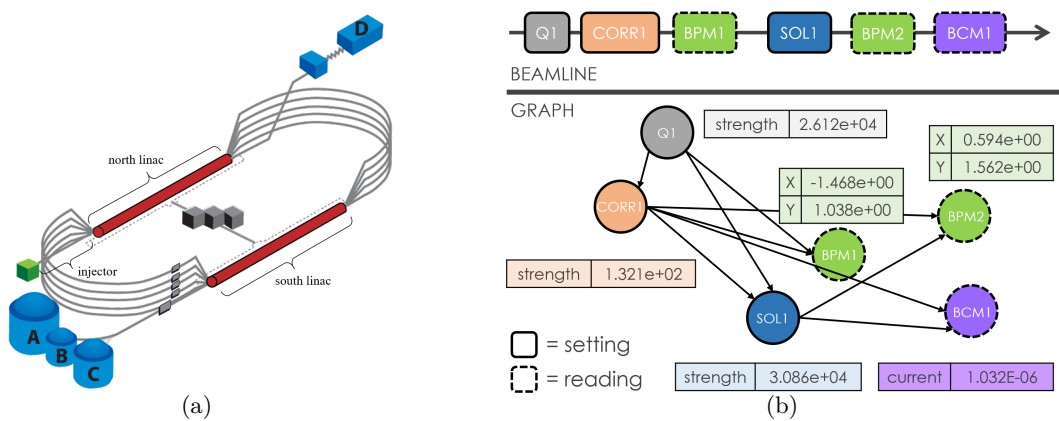

Figure 1: (a) Schematic of the CEBAF accelerator. Electrons are generated in the injector. Multiple passes through the north and south linacs accelerate the beam to multi-GeV energies. The beam is then sent to the four nuclear physics experimental halls (A, B, C, and D). (b) Illustrations that showcase an arbitrary accelerator beamline (top) and our approach for constructing a corresponding graph (bottom). Here, each node represents an individual element, while the node features correspond to the relevant parameters of the respective element. The edges between nodes are determined by a user-defined window size of 2. These edges are directed to reflect the fact that an element cannot impact upstream elements in the beamline.

corresponding to their field strength, BCMs provide beam current measurements, and BPMs include the horizontal and vertical beam positions as well as the sum signal, which is proportional to the beam current in the beamline. The resulting graph is heterogeneous and directed, where heterogeneity arises from the diverse node types, and directionality reflects the causal flow along the beamline.

Edges between nodes are established based on a user-defined "window". In the example in Fig. 1b, a window size of 2 is applied, meaning each ***setting*** node is connected to the two immediate downstream setting nodes, with any intermediate ***reading*** nodes also included. Setting nodes represent elements that operators actively adjust during beam tuning, such as Q1 (the first quadrupole), SOL1 (the first solenoid), and CORR1 (the first corrector), whereas reading nodes passively provide diagnostic feedback (e.g., BPMs and BCMs). The directed edges reflect the unidirectional nature of the beamline, as upstream nodes cannot be influenced by downstream elements. Notably, the window size is adjustable, allowing graph representations to be tailored for specific downstream tasks and beamline characteristics. In a previous work (Wang et al., 2024), the results shows that a window size of 2 achieves the best performance in the classification of beamline configurations, compared to window sizes of 1, 3, and 5. This setting achieves a balance between capturing local interactions and avoiding spurious long-range links. Thus, we follow this work and choose the optimal window size of 2. To facilitate the integration of global beamline parameters, we create a master node connecting to all other nodes, enabling the inclusion of global features such as beam current, environmental conditions (e.g., temperature and humidity), and temporal information. This design allows for a more comprehensive representation of the beamline. For this study, each beamline graph consists of 12 distinct node types, 207 nodes in total, 393 features, and 530 edges (based on a window size of 2), with a master node connecting to all others. Each node type has specific features that can vary significantly in magnitude due to the nature of the physical quantities they represent. To address this, we perform element-wise normalization. For each dataset, we calculate the mean and standard deviation for each node feature and standardize them.

## 2.2 Problem Formulation

In this section, we present a formal definition of the beamline explanation task. Consider a dataset $D$, which is composed of multiple graphs: $D = \{G_1, G_2, \ldots, G_{|D|}\}$. Each individual graph $G$ is characterized by the triplet $G = (\mathcal{V}, \mathcal{E}, \mathcal{X})$. Specifically, $\mathcal{V}$ and $\mathcal{E}$ denote the collection of nodes and edges, respectively. An alternative representation of graph $G$ is through its adjacency matrix $\mathbf{A} \in \mathbb{R}^{n \times n}$, where $n$ corresponds to the number of nodes in the graph. The element $\mathbf{A}_{i,j}$ at the intersection of the $i$-th row and $j$-th column is set to 1 if an edge exists between node $i$ and node $j$; otherwise, $\mathbf{A}_{i,j} = 0$. Given that beamline graphs

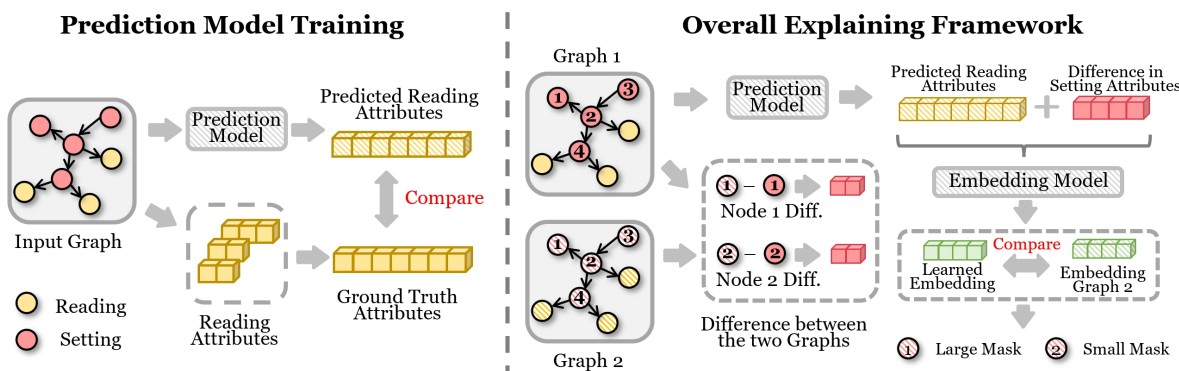

Figure 2: The overall framework. On the left-hand side, we illustrate the training process of the prediction model, which predicts the reading attributes, given the setting attributes of an input beamline graph. On the right-hand side, the explanation framework takes two beamline graphs as input and computes the difference in setting node features between them. Then the framework optimizes the node mask to learn the importance.

are inherently heterogeneous, we introduce a node type mapping function $\tau : \mathcal{V} \to \mathcal{T}$, where $\mathcal{T}$ denotes the set of all node types. With this mapping function, the type of node $v_i$ is given by $\tau(i)$. Furthermore, node features are encoded as $\mathcal{X} = \{\mathbf{x}_1, \mathbf{x}_2, \ldots, \mathbf{x}_n\}$, where $\mathbf{x}_i \in \mathbb{R}^{d_{\tau(i)}}$ represents the feature vector of node $v_i$, and $d_{\tau(i)}$ corresponds to the dimensionality of features specific to the node type $\tau(i)$.

The primary goal is to develop an explanation-driven framework that identifies and ranks the most influential nodes responsible for differences between two beamline graphs. Formally, given a pair of graphs $G_{\text{init}} = (\mathcal{V}, \mathcal{E}, \mathcal{X}^{(1)})$ and $G_{\text{end}} = (\mathcal{V}, \mathcal{E}, \mathcal{X}^{(2)})$, where both share the same topology but differ in node features due to changes in a subset of setting nodes $\mathcal{V}_c \subset \mathcal{V}$, the objective is to compute a ranking over $\mathcal{V}_c$ that reflects the importance of each node in contributing to the observed difference between $G_{\text{init}}$ and $G_{\text{end}}$. This task enables interpretable analysis of how changes of specific elements can propagate through the beamline graph, offering practical guidance for operators during beam tuning.

## 3 Methodology

The primary motivation is to improve accelerator operators' physical intuition about how specific setting adjustments impact broader beamline configurations, thus improving operational efficiency. Accelerator tuning, the process of converging to a desired operational point, is inherently iterative and often time-consuming. By providing insights into the relationship between setting changes and embedding movements, we not only explain the behavior of the system, but also offer practical guidance to recover desired operational states. We seek to determine the most significant setting nodes that account for embedding movement between two points in the learned representation space. This formulation of explainability serves a practical purpose in accelerator operations.

Consider a common scenario: A beamline operating at optimal performance gradually degrades over several days as different operators make adjustments in response to beam variations or environmental changes. While general guidelines exist for injector tuning, the specific implementation varies among operators. In such cases, identifying the primary setting changes relative to a known stable operating point can provide valuable guidance for state recovery.

Determining these important nodes presents significant challenges. The heterogeneous nature of the accelerator system means that the magnitude of setting changes alone is insufficient to determine node importance. For instance, an equivalent normalized change in a quadrupole setting versus an accelerating cavity phase will have substantially different effects on the beam. Moreover, the spatial location of the modified node within the beamline critically influences its downstream effects. This complexity requires a more sophisticated approach to ranking the significance of setting changes.

**Workflow.** Our framework follows a structured workflow consisting of three main components: an *embedding model*, a *prediction model*, and an *explanation model*, as illustrated in Fig. 2. Utilizing the representations learned by the embedding model and the predicted reading attributes from the prediction model, the explanation model identifies the most influential setting nodes contributing to changes between any two beamline graphs. By integrating these components into a comprehensive framework, we aim to enable the explanation for the scientific domain of particle accelerators with unique constraints. Unlike generic contrastive learning, our alignment loss preserves operational geometry between setting nodes, which is crucial for interpretable state transitions in accelerators.

## 3.1 Embedding Model

This section presents our approach to learning graph embeddings for CEBAF injector beamline graphs while accounting for node type diversity. The heterogeneous nature of the accelerator system presents a significant challenge for learning the embeddings, as the magnitude of setting changes alone is insufficient to determine node importance. Therefore, for the embedding model, we introduce a heterogeneous graph convolution method to encode beamline graphs (Hu et al., 2019; Yang et al., 2023; Zhao et al., 2021). This approach addresses variations in feature distributions across different node types by transforming them into a unified latent space through linear layers.

Formally, given a beamline graph $G = (\mathcal{V}, \mathcal{E}, \mathcal{X})$, we define the adjacency matrix with self-connections as $\mathbf{A}' = \mathbf{A} + \mathbf{I}$, where $\mathbf{I}$ is the identity matrix of size $n$, with $n = |\mathcal{V}|$. The degree matrix $\mathbf{M}$ is then given by $\mathbf{M}_{ii} = \sum_{j=1}^{n} \mathbf{A}_{i,j}$. Generally, the propagation rule at each layer in the standard graph convolution operation (Kipf & Welling, 2017; Xu et al., 2019; Hamilton et al., 2017a;b) for homogeneous graphs can be formulated as:

$$\mathbf{h}_v^{(l+1)} = \text{MP}\left(\{\mathbf{h}_u^{(l)} \mid u \in \mathcal{N}(v)\}, \mathbf{h}_v^{(l)}\right), \tag{1}$$

where $\text{MP}(\cdot)$ denotes the message-passing function that can aggregate representations from a set of nodes and output a representation for a specific node. Moreover, $\mathbf{h}_v^{(l+1)}$ denotes the representation of node $v$ at layer $l + 1$. $\mathbf{h}_v^{(0)}$ corresponds to the initial node features of $v$. $\mathcal{N}_v$ denotes the set of neighboring nodes of $v$.

To take into account the heterogeneity of node types and varying feature dimensions, we employ a heterogeneous graph convolution strategy (Hu et al., 2019; Liu et al., 2020; Chang et al., 2015; Hu et al., 2020), allowing integration of multi-type node information. We first project the node features of different node types into a common latent space:

$$\mathbf{h}_v^{(0)} = \mathbf{x}_v \cdot \mathbf{W}_{\tau(v)} + \mathbf{b}_{\tau(v)}, \quad \text{where} \quad \tau(v) \in \mathcal{T}. \tag{2}$$

Here $\mathbf{W}_{\tau(v)} \in \mathbb{R}^{d_{\tau(v)} \times d^{(0)}}$ and $\mathbf{b}_{\tau(v)} \in \mathbb{R}^{d^{(0)}}$ are projection parameters for node type $\tau(v)$ of node $v$. Here, $d_{\tau(v)}$ denotes the input feature dimension for nodes of type $\tau(v)$, and $d^{(0)}$ is the dimension of layer 0. $\mathcal{T}$ is the set of node types. Given this transformation, heterogeneous graph convolution is performed as:

$$\mathbf{H}^{(l+1)} = \text{ReLU}\left(\sum_{\tau \in \mathcal{T}} \tilde{\mathbf{A}}_\tau \cdot \left(\mathbf{H}_\tau^{(l)} \cdot \mathbf{W}_\tau^{(l)} + \mathbf{b}_\tau^{(l)}\right)\right), \tag{3}$$

where $\tilde{\mathbf{A}} = \mathbf{A}'\mathbf{M}^{-1}$ is the normalized adjacency matrix, and $\mathbf{W}^{(l)}$ and $\mathbf{b}^{(l)}$ are the trainable parameters for layer $l$. $\mathbf{W}_\tau^{(l)}$ and $\mathbf{b}_\tau^{(l)}$ are layer-specific parameters for node type $\tau$. Unlike standard GCN, we assign distinct parameters to different node types, ensuring feature transformations into a shared latent space. Additionally, $\tilde{\mathbf{A}}_\tau$ represents a submatrix of $\tilde{\mathbf{A}}$, encoding connections between nodes and their neighbors of type $\tau$.

**Self-Supervised Optimization for Representation Learning.** We propose a self-supervised optimization strategy that encourages alignment between node features and their learned embeddings for more comprehensive representation learning. Specifically, we introduce an alignment loss, which aims to preserve the pairwise distances between nodes in the original feature space within the embedding space. This encourages the embedding space to maintain the structural geometry of the input features and mitigates potential distortions introduced by heterogeneous transformations. Given a pair of node feature vectors $\mathbf{x}_1, \mathbf{x}_2$ and

their corresponding learned embeddings $\mathbf{h}_1, \mathbf{h}_2$, we define the self-supervised alignment loss as:

$$\mathcal{L} = |D(\mathbf{x}_1, \mathbf{x}_2) - D(\mathbf{h}_1, \mathbf{h}_2)|, \quad \text{where} \quad D(\mathbf{x}_1, \mathbf{x}_2) = \sqrt{\sum_{i=1}^{n} (\tilde{x}_{1,i} - \tilde{x}_{2,i})^2}. \tag{4}$$

Here $D(\cdot, \cdot)$ represents the Euclidean distance between standardized feature vectors, with standardization given by: $\tilde{x}_{1,i} = (\mathbf{x}_{1,i} - \mu(\mathbf{x}_1))/\sigma(\mathbf{x}_1)$. This ensures distances in the input feature space are appropriately aligned with distances in the embedding space, promoting consistency in graph representations. This loss aims to explicitly align distances in feature space with those in the latent embedding space, ensuring that operationally meaningful differences (e.g., in quadrupole strengths or cavity phases) are preserved. The embedding space, therefore, maintains a geometry consistent with the input features while also leveraging graph connectivity to capture causal dependencies. This alignment guarantees that minimizing latent distance corresponds to recovering a physically plausible state. An example of nearly 52,000 learned representations after optimization are illustrated in Fig. 3, which reflects the CEBAF operational run in 2022.

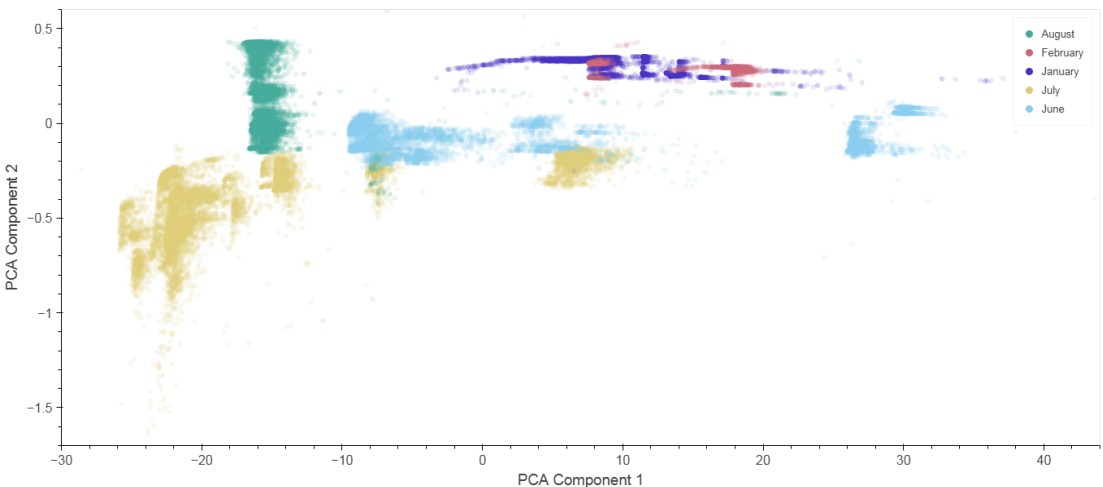

Figure 3: Embeddings representing CEBAF injector operations in 2022. Markers are color-coded by month. Note that there was a scheduled accelerator down during March, April, and May.

## 3.2 Prediction Model

This subsection presents the training details of our prediction model (as shown in the left-hand side of Fig. 2). The purpose of this model is to estimate the features of reading nodes based on the features of setting nodes. This design aligns with the physical constraints of real-world accelerator systems: operators can directly manipulate the values of setting nodes (e.g., magnet strengths, RF phases), but not the values of reading nodes, which passively reflect system diagnostics and feedback.

To capture the relationship between these two types of nodes, we train a dedicated prediction model that learns to approximate the effect of setting adjustments on the readings. The prediction task involves learning a mapping from setting node features to reading node features, which can be formalized as:

$$\widehat{\mathcal{X}}^{\text{read}} = f_\theta(\mathcal{V}_{\text{set}}, \mathcal{E}_{\text{set}}, \mathcal{X}_{\text{set}}), \text{ where } \mathcal{E}_{\text{set}} = \{(v_i, v_j)|v_i, v_j \in \mathcal{V}_{\text{set}}\}, \quad \mathcal{X}_{\text{set}} = \{\mathbf{x}_v|v \in \mathcal{V}_{\text{set}}\} \tag{5}$$

where $f_\theta$ is a graph neural network parameterized by $\theta$, $\mathcal{V}_{\text{set}}$ denotes the set of setting nodes, $\mathcal{E}_{\text{set}} = \{(v_i, v_j) \mid v_i, v_j \in \mathcal{V}_{\text{set}}\}$ is the set of edges among setting nodes, and $\mathcal{X}_{\text{set}} = \{\mathbf{x}_v \mid v \in \mathcal{V}_{\text{set}}\}$ represents the corresponding node features. The model is trained to minimize the mean squared error (MSE) between the predicted and actual reading node features:

$$\widehat{\theta} = \operatorname*{argmin}_{\theta} \frac{1}{|\mathcal{V}|} \sum_{i \in \mathcal{V}} \left\| \widehat{\mathcal{X}}_i^{\text{read}} - \mathcal{X}_i^{\text{read}} \right\|^2. \tag{6}$$

This objective ensures that the model learns an accurate functional approximation of how beamline readings respond to configuration changes.

## 3.3 Explanation Model

Our explanation model interprets the relationships between setting attributes and reading attributes in a self-supervised manner (as shown in the right-hand side of Fig. 2). We first train a prediction model that learns to infer reading attributes based on setting attributes.

Given a trained GNN model $\Phi$, we aim to identify the most important setting nodes whose features and connectivity explain the change between two beamline graphs. To achieve this, we seek to learn a continuous node mask that highlights which nodes contribute most to the observed shift in representation between two beamline graphs.

Let $G = (\mathcal{V}, \mathcal{E}, \mathcal{X})$ be a beamline graph with node features $\mathcal{X}$, and let $H = g(\mathcal{V}, \mathcal{E}, \mathcal{X}) \in \mathbb{R}^d$ denote the dense representation produced by the GNN. Given a pair of graphs $G_{\text{init}}$ and $G_{\text{end}}$ where certain setting nodes have changed, we denote their respective representations as $H^{(1)}$ and $H^{(2)}$. Our goal is to explain the transition between these two states by identifying the explanation graph $G_S$, defined by the node mask $\mathbf{m} = \{m_1, m_2, \ldots, m_n\} \in [0,1]^n$, where $n$ is the number of changed setting nodes. With the explanation graph $G_S$, we aim to provide insights of nodes whose features and connectivity are most informative about the latent transition from $H^{(1)}$ to $H^{(2)}$.

**Optimization via Mutual Information.** To effectively learn the masks to constitute $G_S$, we propose to utilize the concept of mutual information for optimization. Particularly, we aim to maximize the mutual information between the explanation graph $G_S$ and the target representation shift:

$$\max_{G_S} \ \text{MI}(H, G_S) = \mathcal{H}(H) - \mathcal{H}(H \mid G = G_S, \mathcal{X} = \mathcal{X}_S), \tag{7}$$

where $\mathcal{X}_S$ denotes the masked input features corresponding to $G_S$, and $\mathcal{H}(\cdot)$ denotes differential entropy. Notably, $\mathcal{H}(H)$ is fixed for a trained representation model. Thus, maximizing the mutual information is equivalent to minimizing conditional entropy:

$$\min_{G_S} \ \mathcal{H}(H \mid G = G_S, \mathcal{X} = \mathcal{X}_S) = -\mathbb{E}_{H \mid G_S} \left[ \log p(H \mid G_S, \mathcal{X}_S) \right]. \tag{8}$$

In practice, we model this process through reconstruction: we seek a masked input of the beamline graph that produces a representation as close as possible to the true representation $H^{(2)}$. The explanation is thus formulated as the mask that minimizes the distance between the masked-output representation and $H^{(2)}$.

**Variational Mask Optimization.** To enable gradient-based optimization, we use a continuous relaxation of the binary node mask $\mathbf{m} \in [0,1]^n$. This mask is applied individually to each setting node (on its features), such that the node features of the masked graph becomes $\mathcal{X}(\mathbf{m})$. The masked representation is then computed as:

$$\widehat{H}^{(2)} = \Phi(\mathcal{V}, \mathcal{E}, \mathcal{X}(\mathbf{m})), \quad \text{where} \quad \mathcal{X}(\mathbf{m}) = \mathcal{X}_{\text{set}}(\mathbf{m}) \cup \mathcal{X}_{\text{read}}(\mathbf{m}). \tag{9}$$

Here $\mathcal{X}(\mathbf{m})$ denotes the new node features. Particularly, we calculate $\mathcal{X}(\mathbf{m})$ separately for setting nodes and reading nodes as follows. For setting nodes, we consider the difference between the two beamline graphs $G_{\text{init}}$ and $G_{\text{end}}$. If a specific setting node changes between these two graphs, we will learn a node mask for it. In this way, the features of $G_{\text{init}}$ added with the node mask multiplied by the change will become the new features. For setting nodes that are not changed, their features remain unchanged. The calculation is as follows:

$$\mathcal{X}_{\text{set}}(\mathbf{m}) = \{\mathbf{x}_i^{(1)} + \delta_i \cdot (\mathbf{x}_i^{(2)} - \mathbf{x}_i^{(1)}) \mid v_i \in \mathcal{V}_{\text{set}}\}, \quad \text{where} \quad \delta_i = \begin{cases} m_i & \text{if } v_i \in \mathcal{V}_{\text{change}}, \\ 0 & \text{if } v_i \notin \mathcal{V}_{\text{change}}. \end{cases} \tag{10}$$

Here $\mathcal{V}_{\text{change}}$ denotes the set of setting nodes changed between the two graphs. For the reading node features, it is inappropriate to use the same adding strategy, as the changes of reading node features are dependent

on the collective impact of multiple setting nodes. Therefore, we leverage the trained prediction model $f_\theta$ to obtain the predicted reading node feature:

$$\mathcal{X}_{\mathrm{read}}(\mathbf{m}) = f_\theta(\mathcal{V}_{\mathrm{set}}, \mathcal{E}_{\mathrm{set}}, \mathcal{X}_{\mathrm{set}}(\mathbf{m})). \tag{11}$$

The predicted reading node features, together with the new setting node features, will constitute the features for $G_S$. To optimize the node masks, i.e., $\mathbf{m}$, we propose the following objective to obtain the optimal node masks $\widehat{\mathbf{m}}$ as follows:

$$\widehat{\mathbf{m}} = \arg\min \|\widehat{H}^{(2)} - H^{(2)}\|^2 + \lambda\|\mathbf{m}\|_1 \tag{12}$$

where $\lambda$ is a regularization hyperparameter that encourages sparsity in the learned mask, thereby improving the interpretability of the explanation. The optimization objective is to minimize $\|\widehat{H}^{(2)} - H^{(2)}\|$, ensuring that the modified graph remains as similar as possible to the target graph in the latent space. This enables the model to effectively capture the underlying structure and reasoning behind the transformations in the dataset.

By optimizing this loss, we obtain a soft mask $\mathbf{m}$ that highlights the most influential nodes driving the representation shift between two graphs. The result is an interpretable explanation in terms of node importance, revealing which setting changes contributed the most significantly to the transition. This explanation supports human operators in understanding beamline behavior and facilitates more effective control in real-world accelerator settings.

### 3.4 Establishing the Ground Truth

Particle accelerator data lacks ground truth annotations for identifying important nodes in explanations, making it difficult to evaluate our explanation framework's performance and compare it fairly with existing baselines. To address this challenge, we propose a statistical analysis approach to approximate the ground truth by quantifying node importance based on their individual contributions to changes between two beamline graphs. This serves as a principled way to construct a proxy for ground truth, enabling more meaningful evaluation. The core idea is to iterate over all possible permutations of the changed setting nodes, as

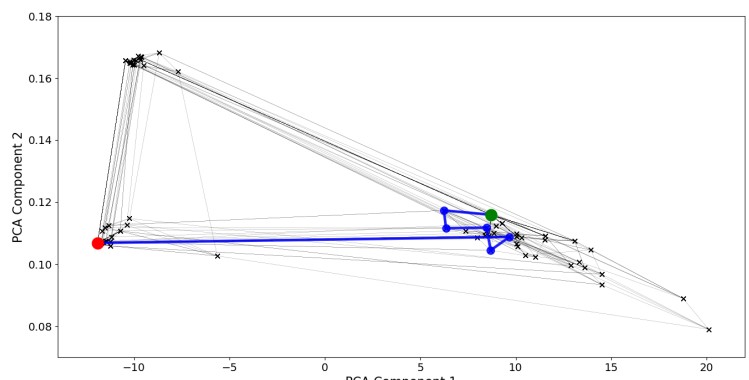

Figure 4: Visualizing 720 possible trajectories between the initial (green) and end (red) configuration. The blue line represents the actual path taken in the experiment.

the order in which node changes are applied can also influence the resulting graph difference. This is illustrated in Fig. 4. The blue trace represents the actual trajectory in the low-dimensional space while moving between an initial (green) and end (red) injector configuration during a beam study at CEBAF. The black traces represent the 719 other possible trajectories if we consider all permutations of the six setting node changes. By evaluating every permutation, we can mitigate the effect of ordering bias and assess each node's impact more fairly. For each permutation, we compute the marginal contribution of each node by measuring how much it reduces the MAE (Mean Absolute Error) between the readings of two beamline graphs. The nodes are then ranked based on their average contribution across all permutations.

This exhaustive approach becomes computationally infeasible when the number of changed nodes is large (e.g., $24! \approx 6 \times 10^{23}$ permutations). To make this process tractable, we adopt a more efficient strategy: first, we identify the top-$k$ most influential nodes based on a heuristic ranking, and then we enumerate all $k!$ permutations of this reduced subset. In the experiments, we set $k = 4$. This significantly reduces the computational burden while preserving the fidelity of the importance estimates.

**Important Node Identification.** We analyze the impact of individual node modifications within a graph, considering two distinct cases, each represented as a graph with a specific number of nodes. We define one

of these graphs as the *starting case* and systematically apply individual node modifications to observe their effects. The key objective is to determine which nodes contribute most significantly to changes in the overall graph behavior.

Formally, given an initial graph $G_{\text{init}} = (\mathcal{V}, \mathcal{E}, \mathcal{X})$ and a set of changed nodes $\mathcal{V}_{\text{change}}$, we iteratively apply each modification $v_i \in \mathcal{V}_{\text{change}}$ to generate a modified graph $G_{\text{mod}}(v_i)$. The difference between the original and modified graphs is quantified using the change in prediction error, defined as:

$$\Delta D(v_i) = |f_\theta(G_{\text{mod}}(v_i)) - f_\theta(G_{\text{init}})|, \tag{13}$$

where $\Delta D(v_i)$ denotes the prediction error of the model when modification $v_i$ is applied.

We rank all changed based on $\Delta D(v_i)$ and identify the top $k$ nodes that induce the most significant changes in graph behavior. The selected nodes, $\mathcal{V}_{\text{top},k}$, are as follows:

$$\mathcal{V}_{\text{top},k} = \operatorname*{argmax}_{V'} \sum_{v_i \in V'} \Delta D(v_i), \quad \text{where} \quad V' \subset \mathcal{V}_{\text{change}}, \quad |V'| = k. \tag{14}$$

**Ground Truth Establishment via All Permutations.** To evaluate the impact of these nodes, we compute all possible permutations of these $k$ nodes and analyze their contributions by running the prediction model across all $k!$ possible orderings. In this way, the specific node that causes the largest MAE reduction in readings from $G_{\text{init}}$ to $G_{\text{end}}$ is considered the most important node in this permutation. Ultimately, the nodes that have the largest proportion of being the most important nodes across all permutations will be considered the most influential nodes. This allows us to systematically assess the influence of node interactions. Notably, the established ground truth is based on marginal contribution to prediction error, not directly on the masks learned by the explanation model. Thus, while both use the prediction model, the explanation model optimizes in the embedding space, whereas the established ground truth evaluates marginal error reduction. Although the real ground truth is not available at the current stage, future directions will include collaborating with CEBAF operators for validation and integrating expert feedback.

**Order-Dependence.** Notably, our establishment of ground truth naturally incorporates the order-dependence effect, which is explicitly considered among the most influential $k$ nodes. Rather than simply averaging over all permutations, our method applies a principled filter that balances computational feasibility with fairness in evaluating node importance. Importantly, this design also reflects the practical reality of accelerator operations, where operators often make iterative adjustments, making it difficult to define a strict sequential ground truth.

**Case Study.** In Fig. 4, six setting nodes change between the two graphs. In this case, we exhaustively evaluated all 720 possible permutations of the six node changes and compared them with the results produced by our framework. Our framework ranked the nodes in descending order of importance as 6, 1, 2, 4, 5, 3. Notably, both approaches consistently identified the same most influential node: the setting node associated with the last change, which shifted the value of PCA component 1 from 10 to -12. This agreement between the exhaustive permutation analysis and our framework highlights the validity of our explanation method.

## 4 Experiments

To evaluate our proposed framework, we perform experiments on real-world datasets by analyzing heterogeneous graph-based settings and their influence on prediction performance.

### 4.1 Experimental Setup

Our framework is implemented using PyTorch (Paszke et al., 2017), scikit-learn (Pedregosa et al., 2011), and PyTorch Geometric (Fey & Lenssen, 2019). We run the model on a single NVIDIA A6000 GPU with 48GB of memory. The batch size is set to 16 for self-supervised training and 32 for supervised fine-tuning. The model is trained for a total of 200 epochs with a learning rate of 0.001. The hidden size of the GNN models in our framework is set to 16, and we use a 3-layer architecture. The model is optimized using Adam (Kingma & Ba, 2015). We conduct large-scale experiments on the 2022 dataset. Each graph represents a unique case, and we analyze the impact of node modifications across a wide variety of graph structures. For the 2022

dataset, we consider two splits: (1) Jan-Feb and (2) Jun-Jul-Aug. The detailed statistics are provided in Table 1. We choose these two splits because they are closer together in time. We randomly select 250 samples from each of the two splits to constitute the final test set. The empirical results using various evaluation metrics are provided in Table 2. Code is at https://github.com/SongW-SW/CEBAF-Exp.

Table 1: Statistics of two beamline datasets.

| Dataset | # Graphs | # Edges | # Nodes | # Types |
|---|---|---|---|---|
| Jan-Feb | 25,916 | 530 | 207 | 12 |
| Jun-Jul-Aug | 25,801 | | | |

## 4.2 Baselines and Evaluation Metrics

We compare our framework with several state-of-the-art explanation methods for graph-based learning:

- **GNNExplainer** (Ying et al., 2019)**:** A model-agnostic approach that learns node importance by optimizing a mutual information-based objective to explain GNN predictions.

- **PGExplainer** (Luo et al., 2020)**:** A parameterized explainer that learns a probabilistic mask for graph structures to highlight important nodes and edges.

- **Gradient Integration:** An explanation approach that utilizes backpropagated gradients to measure the importance of features in node embeddings, identifying their influence on final predictions.

- **Feature:** A feature-based approach that evaluates the impact of individual node features by analyzing the magnitude of node embeddings in trained graph models.

## 4.3 Evaluation Metrics

For each experiment, we randomly select two graphs $G_1$ and $G_2$ and identify the most important node differences between them. Using our framework, we rank nodes in $\mathcal{V}_{\text{change}}$ with our explanation results. To quantitatively assess the effectiveness of the results, we compute two metrics: (1) Precision@$k$ measures the overlap between the identified important nodes and the ground-truth. (2) Hits@$k$ is a recall-based metric that measures whether at least one of the top-$k$ predicted nodes appears in the ground-truth set. These two metrics can be formulated as above:

$$\text{Precision@}k = \frac{|\mathcal{V}_{\text{pred},k} \cap \mathcal{V}_{\text{top},k}|}{k}, \qquad \text{Hits@}k = \begin{cases} 1, & \text{if } \mathcal{V}_{\text{pred},k} \cap \mathcal{V}_{\text{top},k} \neq \emptyset, \\ 0, & \text{otherwise.} \end{cases} \tag{15}$$

Here $\mathcal{V}_{\text{pred},k}$ represents the top-$k$ nodes predicted as influential, and $\mathcal{V}_{\text{top},k}$ is the ground-truth set of the top-$k$ influential nodes, established in Eq. (14). Notably, unlike Precision@$k$, which quantifies the proportion of correctly identified nodes, Hits@$k$ provides a binary measure of whether at least one correct node was retrieved within the top-$k$ predictions. This metric is particularly useful for evaluating whether a method can successfully identify critical nodes in high-dimensional settings. By considering both Precision@$k$ and Hits@$k$, we aim to provide a more comprehensive understanding of how well different methods identify important nodes in beamline graphs.

## 4.4 Empirical Results on Large Datasets

The results demonstrate the effectiveness of our proposed method compared to various baselines. The key observations are summarized as follows:

- **Superior Performance:** Our approach achieves the highest performance across multiple metrics, particularly in Hits@5 (42.25%) and Hits@10 (44.26%), demonstrating its effectiveness in identifying key nodes within graph structures.

- **Incremental Gains from PGExplainer over GNNExplainer:** PGExplainer slightly outperforms GNNExplainer across all metrics, suggesting that incorporating additional probabilistic explanations enhances interpretability and accuracy.

Table 2: Performance comparison across different methods.

| Method | Hits@1 | Hits@3 | Hits@5 | Hits@10 | Precision@3 | Precision@5 |
|---|---|---|---|---|---|---|
| Feature | 6.35% | 14.20% | 34.05% | 36.82% | 25.49% | 28.21% |
| Gradient Integration | 5.39% | 12.69% | 31.52% | 35.73% | 27.02% | 29.35% |
| GNNExplainer | 8.52% | 16.39% | 37.43% | 41.90% | 27.48% | 32.50% |
| PGExplainer | **9.07**% | 16.49% | 38.34% | 42.58% | 28.25% | 32.77% |
| Ours | 8.76% | **17.84**% | **42.25**% | **44.26**% | **28.55**% | **33.13**% |

Table 3: Performance comparison under different initialization values.

| Method | Hits@1 | Hits@3 | Hits@5 | Hits@10 | Precision@3 | Precision@5 |
|---|---|---|---|---|---|---|
| Ours (Init=0.5) | **8.76%** | **17.84%** | **42.25%** | **44.26%** | **28.55%** | **33.13%** |
| Ours (Init=0.75) | 7.51% | 16.94% | 41.99% | 43.05% | 27.98% | 32.97% |
| Ours (Init=1.0) | 7.13% | 16.27% | 39.82% | 40.85% | 27.91% | 32.87% |
| Ours (Init=0) | 6.56% | 15.62% | 38.92% | 39.66% | 27.47% | 31.70% |

- **Limited Performance of Gradient Integration:** Gradient Integration lags behind other methods, particularly in Hits@1 (5.39%) and Hits@3 (12.69%), indicating that it struggles to accurately highlight the most influential nodes.

- **Feature Approach as a Weak Baseline:** The Feature method consistently shows the lowest results, with Hits@1 at only 6.35%, reaffirming the necessity of advanced explainability methods in graph-based tasks.

- **Precision Trends:** Our method also achieves the highest precision scores, with 28.55% at Precision@3 and 33.13% at Precision@5, highlighting its ability to prioritize correct nodes more effectively.

- **Trade-off in Hits vs. Precision:** While PGExplainer slightly improves over our method in the Hits@1 metric, its Precision@k value is lower than our method, indicating that PGExplainer enhances the ranking of the top several nodes but does not drastically improve the correctness of top selections.

Notably, all values are below 50% under the challenging setting in our experiments, as multiple nodes may plausibly contribute to observed outcomes. Importantly, our method consistently outperforms baseline approaches and demonstrates clear relative effectiveness. In sum, these results validate the effectiveness of our proposed approach, demonstrating superior generalizability and robustness in identifying critical nodes within large-scale graph datasets.

### 4.5 Parameter Sensitivity

We explore the effects of various hyperparameters, specifically analyzing how different initialization values impact model performance. We experiment with three initialization values of weight masks: `Init=0`, `Init=0.5`, and `Init=1.0`, and compare them against the baseline methods. The results in Table 3 illustrate the sensitivity of our method to different initialization values. Key observations include:

- **Optimal Performance at Init=0.5:** Our method achieves the highest results at `Init=0.5`, outperforming all other initialization settings in Hits@k and Precision@k metrics.

- **Decreasing Performance with Init=0 or Init=1.0:** Setting the initialization to 0 or 1.0 results in a drop in performance across all metrics, indicating that an intermediate initialization provides a better balance between model stability and adaptability.

Table 4: Ablation study results showing the impact of different components on performance.

| Method | Hits@1 | Hits@3 | Hits@5 | Hits@10 | Precision@3 | Precision@5 |
|---|---|---|---|---|---|---|
| Ours | **8.76%** | **17.84%** | **42.25%** | **44.26%** | **28.55%** | **33.13%** |
| Ours w/o Graph | 6.68% | 14.91% | 39.06% | 41.58% | 26.80% | 29.79% |
| Ours w/o Emb. | 7.28% | 15.88% | 40.26% | 41.76% | 27.32% | 28.64% |

- **Comparison with Baselines:** Even under suboptimal initialization settings (Init=0 or Init=1.0), our method still surpasses the Feature Value baseline in all metrics, demonstrating robustness.

- **Hits vs. Precision Trade-off:** Although the overall Hits@k performance is the highest for Init=0.5, the Precision@k scores show a smaller variance across different settings, suggesting that precision is less sensitive to initialization values.

These findings highlight the importance of careful hyperparameter selection, particularly in choosing initialization values that optimize both model convergence and prediction accuracy.

### 4.6 Ablation Study

We conduct an ablation study by systematically removing key components and evaluating their impact on performance. Specifically, we consider the following variations:

- **Ours:** The full version of our method incorporating all three models, including the embedding model, the prediction model, and the explanation model.

- **Ours w/o Graph:** A variant where the graph-based embedding model is removed, assessing the impact of graph structures. In this variant, the GNN embedding model is replaced with multiple linear layers.

- **Ours w/o Emb.:** A variant where the optimization loss for the node masks is calculated based on reading attributes instead of learned embeddings. As a result, the embedding model is not used in the computation and optimization of the node masks.

The results in Table 4 highlight the importance of each component in our framework:

- **Impact of Graph-Based Reasoning:** Removing the graph-based embedding model (`Ours w/o Graph`) leads to a noticeable decline in performance across all metrics, particularly in Hits@k scores, demonstrating the crucial role of structured graph information in enhancing predictions.

- **Effect of Removing Embedding Model in Explanation Computation:** Excluding the embedding model (`Ours w/o Emb.`) also results in a performance drop, though it is less pronounced than the removal of graph-based reasoning. This suggests that embeddings contribute significantly to refining node importance but are more effective when combined with graph structures.

- **Overall Contribution of Components:** The full model consistently outperforms both ablated versions, confirming that both graph-based reasoning and embedding-based enhancements are complementary and necessary for optimal performance.

This ablation study underscores the necessity of integrating structured graph reasoning and embedding-based feature refinement to achieve superior reasoning performance in our framework.

## 5 Conclusion

In this work, we presented an explanation-driven framework for optimizing particle accelerator beamline operations through graph-based machine learning. By representing beamline configurations as heterogeneous graphs and applying a novel node-masking approach, our method successfully identifies and ranks the

most influential setting nodes responsible for observed changes in beam behavior. Experimental results on real-world CEBAF injector data demonstrate that our approach outperforms existing explainability methods across multiple metrics, with particular improvements in Hits@5 (42.25%) and Precision@5 (33.13%). Ablation studies further confirm the importance of both graph-based reasoning and embedding-based enhancements in achieving these results. This framework offers a practical tool for accelerator operators, providing interpretable insights that can significantly reduce tuning time and improve operational efficiency. This successful proof-of-concept demonstration motivates its use on more recent data and deployment as an operational tool in the control system. By bridging the gap between complex physical systems and interpretable AI, our work represents an important step toward more explainable and efficient accelerator operations, with potential applications extending to other scientific instruments and industrial systems, where understanding cause-effect relationships in high-dimensional settings is crucial.

## Acknowledgements

This material is based on work supported by the U.S. Department of Energy, Office of Science, Office of Nuclear Physics under Contract No. DE-AC05-06OR23177.

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
