# OpenReview forum: "Explainable Graph Learning for Particle Accelerator Operations"
_TMLR — Accepted by TMLR_

### Review · Reviewer_L9ir · 2025-06-21

**Summary Of Contributions:**

Motivated by the time-consuming and challenging task of beam tuning at the Continuous Electron Beam Accelerator Facility (CEBAF), this paper aims to devlop an explanation-driven framework to identify and rank influential "setting nodes" responsible for feature differences between two beamline graphs. Experiments were conducted to evaluate the proposed methods.

**Audience:**

No

**Broader Impact Concerns:**

Nil

**Claims And Evidence:**

No

**Requested Changes:**

- Please address my concerns in the weaknesses section
- Question: Why is it reasonable to treat node features across the heterogenous types in the same latent space? For example, if one node controls temperature and another controls pressure, why should "high temperature" and "high pressure" features should have the same latent representation? Are setting nodes "exchangeable" in some sense?
- Question: How should I interpret the numbers in the evaluation? Everything is below 50%. Is that supposed to be good?
- Typo: Missing space between "graphG" in the second paragraph of Section 3.1

**Strengths And Weaknesses:**

# Strengths

- The problem is well motivated with a concrete application domain in mind
- I am not an expert in beam tuning and what has been done, but it seems interesting that the authors tried to use modern machine learning tools such as GNNs to tackle this problem

# Weaknesses

- The proposed framework is largely a combination of well-known methods applied to a new domain and the methodological contributions appear incremental at best. For instance, the section on "self-supervised optimization for representation" can be viewed as a variation/instance of the "self-supervised contrastive learning", which is well-studied but not referenced/compared against in the paper.
- The approach proposed by the paper seems to have limited generalizability beyond the specific problem studied here. Furthermore, despite having explainability as a goal, there was no demonstration that the outputs are truly understandable or usable by real operators.
- I am concerned with the evaluation process. From equation (13) in Section 3.4, we see that the pseudo ground truth is generated using the same $f_\theta$ that is being used by the method. How is this a fair evaluation? I also suspect that the proposed approach for generating pseudo ground truth based on combinatorial simulations will be prone by spurious correlations and does not offer a strong validation signal. Lastly, there is no human or domain-expert validation to confirm whether the identified nodes are truly important.

---

> ### Author Response · Authors · 2025-09-02
> **Response to Reviewer L9ir**
>
> Dear Reviewer L9ir,
>
>
> &nbsp;
>
>
> Thank you for your diligent review and insightful recommendations. We have carefully considered your feedback and have addressed your concerns in the following manner:
>
>
> &nbsp;
>
>
> ### **Concern 1: Incremental contributions**
>
> **Comment:** Framework appears to combine existing methods; self-supervised loss resembles contrastive learning but not referenced.
>
> **Response:**
> We respectfully clarify that while our components build on established ideas (e.g., heterogeneous GNNs, self-supervised alignment), the novelty lies in their integration for a new scientific domain with unique constraints. Unlike generic contrastive learning, our alignment loss preserves operational geometry between setting nodes, which is crucial for interpretable state transitions in accelerators. We will revise Section 3.1 to more explicitly relate our method to contrastive learning and highlight this adaptation.
>
> ---
>
> &nbsp;
>
>
> ### **Concern 2: Generalizability beyond CEBAF**
>
> **Response:**
> We acknowledge that our evaluation is focused on CEBAF injector data. However, the graph-based representation and explanation framework are general to systems where tunable controls (settings) influence observable diagnostics (readings). Examples include other accelerators, fusion reactors, or industrial control systems. We will strengthen the discussion of generalizability in the conclusion.
>
> ---
>
> &nbsp;
>
>
> ### **Concern 3: Fairness of evaluation / proxy ground truth**
>
> **Comment:** Ground truth generated using same prediction model may introduce circularity; possible spurious correlations; no human validation.
>
> **Response:**
> We appreciate this important concern. We would like to clarify that the proxy ground truth is based on marginal contribution to prediction error, not directly on the explainer’s learned masks. Thus, while both use the prediction model, the explainer optimizes in embedding space, whereas the proxy evaluates marginal error reduction. This constitutes two different mechanisms. To further address this, we will emphasize this distinction and explicitly note limitations in the revised version. Moreover, we are collaborating with CEBAF operators for future validation and plan to integrate expert feedback in follow-up studies.
>
> ---
>
> &nbsp;
>
>
> ### **Concern 4: Node feature latent space consistency**
>
> **Question:** Why is it reasonable to treat heterogeneous features (e.g., temperature vs pressure) in the same latent space?
>
> **Response:**
> Our heterogeneous graph convolution projects each node type through a type-specific transformation before mapping to the shared latent space (Eq. 2). This ensures that raw features with different scales and semantics are normalized and appropriately transformed, while still enabling cross-type interactions that reflect the causal flow of the beamline. We will make this clearer in Section 3.1.
>
> ---
>
> &nbsp;
>
>
> ### **Concern 5: Evaluation metric interpretation (<50% numbers)**
>
> **Response:**
> We appreciate this request for clarification. Hits\@k and Precision\@k are recall/precision metrics, and values below 50% are expected in this challenging setting, as multiple nodes may plausibly contribute to observed outcomes. Importantly, our method consistently outperforms baseline approaches (e.g., Hits\@5 of 42.25% vs. 37.43% for GNNExplainer), demonstrating clear relative effectiveness. We also note that this represents novel work, and there is currently no established benchmark or standard metric for evaluating node importance in accelerator beamline explanations. Our evaluation therefore compares against state-of-the-art explanation methods as reasonable baselines, highlighting the relative gains achieved by our approach. We will expand Section 4.3 to explicitly provide this context.
>
> ---
>
> &nbsp;
>
>
> ### **Typo**
>
> **Response:**
> Thank you for pointing this out. We will correct the missing space between *“graphG”* in Section 3.1.
>
> ---
>
> &nbsp;
>
>
> We sincerely hope our response could address your concerns. Looking forward to your reply!
>
> Best,
> Authors

---

### Review · Reviewer_GpC3 · 2025-07-10

**Summary Of Contributions:**

This paper introduces an explanation-driven framework aimed at facilitating interpretability and efficient operational control of particle accelerator beamlines. Specifically, the authors model particle accelerators as heterogeneous graphs, distinguishing between actively adjustable nodes (setting nodes) and passive diagnostic nodes (reading nodes). The proposed approach employs a GNN-based embedding model to encode graph structures and an explanation model that learns node-level importance via trainable node masks. This enables the identification of critical nodes responsible for beamline configuration changes. Experiments conducted on real-world data demonstrate that the method successfully identifies influential setting adjustments, significantly improving interpretability and potentially reducing operational overhead.

**Audience:**

Yes

**Broader Impact Concerns:**

Currently, the paper does not explicitly discuss broader ethical or societal impacts. While direct ethical concerns may be minimal, the authors should briefly address implications regarding operator trust and reliance on automated explanations in safety-critical decisions, potentially adding a concise Broader Impact Statement.

**Claims And Evidence:**

Yes

**Requested Changes:**

***Critical:*
1. Clarify Figure 1 (b): Figure 1(b) has two instances labeled "BPM1." Authors should clarify this potential typo; one likely should be "BPM2."
1. Define "Q1" Node in Figure 1 (b): Explicitly describe the "Q1" node to clarify its specific role or type within the beamline configuration.
1. Justify Window Size Selection: Explicitly justify the choice of the window size of 2 by reporting empirical results from experiments with alternative window sizes or clearly referencing and summarizing findings from Wang et al., 2024.
1. Explicit Discussion of Figures 2, 3, and 4: Clearly connect these figures to corresponding manuscript sections:
    - Figure 2: Clearly map figure elements to methodological steps described.
    - Figure 3: Explain the embeddings’ implications explicitly—what does this visualization reveal about model performance?
    - Figure 4: Clearly explain the visualization of possible trajectories and its importance in understanding or validating the explanation methodology.
1. Clarify Ground Truth Approximation Method: Provide additional justification or theoretical backing for the validity and potential biases or limitations of the permutation-based ground-truth approximation.

**Non-critical**
1. Scalability Analysis: Briefly discuss computational complexity and scalability, particularly for real-time scenarios.
1. Visualization Enhancements: Include additional visual examples, such as node importance heatmaps, to further enhance interpretability and reader understanding.

**Strengths And Weaknesses:**

**Strength:**
1. Clearly distinguishes adjustable (setting) and observable (reading) nodes, reflecting practical constraints.
1. Combines heterogeneous GNNs with interpretability, addressing a practical, real-world operational optimization problem.
1. Real-world applicability that could significantly improve operator efficiency and reduce downtime.

**Weakness:**
1. Absence of true ground truth explained via permutation-based statistical analysis, needing further validation or theoretical justification.
1. High computational load of exhaustive permutation evaluation could limit practical applicability.
1. Lack of Explicit Discussion of Figures 2, 3, and 4.

---

> ### Author Response · Authors · 2025-09-02
> **Response to Reviewer GpC3**
>
> Dear Reviewer UB34,
>
> &nbsp;
>
>
> Thank you for your detailed review, suggestions, and critical feedback from both the previous and current rounds. Your insightful comments have significantly contributed to the improvement of our paper.
>
>
> &nbsp;
>
>
>
> ### **Critical Comments**
>
> 1. **Figure 1(b) duplicate label (BPM1 twice)**
>    **Response:** Thank you for catching this typo. The second instance should indeed be *BPM2*. We will correct this in the figure and caption.
>
> 2. **Definition of “Q1” node**
>    **Response:** Q1 refers to the first quadrupole magnet in the injector beamline. We will explicitly define this in Section 2.1 and in the figure caption.
>
> 3. **Justification of window size (set to 2)**
>    **Response:** Prior work (Wang et al., 2024) shows that window size 2 achieves the best performance in the classification of beamline configurations, compared to window sizes of 1, 3, and 5. This setting achieves a balance between capturing local interactions and avoiding spurious long-range links. We will add a more detailed discussion in this subsection in the revised version.
>
> 4. **Explicit discussion of Figures 2, 3, and 4**
>    **Response:** We agree this is important. We will revise the text to:
>
>    * Map **Figure 2** directly to the steps of the embedding, prediction, and explanation workflow.
>    * Clarify **Figure 3** by explaining that embeddings cluster operational states over time, providing evidence that the learned latent space captures meaningful dynamics.
>    * Explain **Figure 4** by emphasizing its role in motivating the permutation-based ground truth, and why averaging trajectories creates a fair evaluation baseline.
>
> 5. **Ground truth approximation validity**
>    **Response:** We will explicitly discuss biases and limitations of the permutation-based proxy in the revised version. Particularly, we adopt a two-stage strategy: (1) We first evaluate the effect of each setting node independently, ranking them by their marginal contribution to lowering the prediction error (MAE). (2) We then take the top-K nodes and exhaustively enumerate all permutations within this reduced set.
>
> In this way, order-dependence is explicitly considered among the most influential nodes, while still keeping the computation tractable. Rather than simply averaging over all permutations, our method applies a principled filter that balances computational feasibility with fairness in evaluating node importance. Importantly, this design also reflects the practical reality of accelerator operations, where operators often make iterative and non-standardized adjustments, making it difficult to define a strict sequential ground truth. We will revise Section 3.4 to more clearly describe this process and adjust the language around “average contributions” for clarity.
>
> ---
>
> &nbsp;
>
>
>
> ### **Non-Critical Suggestions**
>
> * **Scalability Analysis:**
>   We will include a short complexity discussion in the revised version. Particularly, exhaustive permutations scale factorially, but our top-$K$ subset strategy (Sec. 3.4) ensures tractability as we limit the number of nodes selected for the exhaustive permutations to $K$. For real-time use, approximation or sampling strategies can also further reduce computational load.
>
> * **Visualization Enhancements:**
>   In the revised version, we will add a node-importance heatmap to supplement quantitative results, providing more intuitive interpretability for practitioners.
>
> * **Broader Impact Statement:**
>   We will add a brief note on operator trust in automated explanations, emphasizing that our method is intended as a decision-support tool, not a replacement for expert judgment in safety-critical operations.
>
> ---
>
> &nbsp;
>
> We sincerely hope our response could address your concerns. Looking forward to your reply!
>
> Best,
> Authors

---

### Review · Reviewer_B3WP · 2025-08-19

**Summary Of Contributions:**

This paper presents a novel, explanation-driven framework using graph-based machine learning to improve the operational efficiency of particle accelerators. Focusing on the injector beamline at the Continuous Electron Beam Accelerator Facility (CEBAF), the authors model the system as a heterogeneous graph to identify the most influential machine settings that cause changes in the electron beam's behavior.

The primary contribution of this work is a three-part framework designed to provide actionable insights for human operators during beam tuning. The contributions are:

1. The authors model the accelerator beamline as a heterogeneous, directed graph. Nodes are categorized as setting nodes (e.g., magnets, correctors), which operators can adjust, and reading nodes (e.g., beam position monitors), which provide diagnostic feedback. This structure effectively captures the physical components and the directional flow of the beamline.

2. The core of the work is an explanation model that identifies which setting adjustments are most responsible for a change between two different beamline states (e.g., from an optimal state to a drifted state). This is achieved through a workflow that includes: (a) An embedding model that learns to represent the entire beamline graph in a low-dimensional space using heterogeneous graph convolutions. (b) A prediction model that is trained to estimate the values of reading nodes based solely on the values of setting nodes, mimicking the real-world cause-and-effect relationship. (c) An explanation model that optimizes a "node mask" to identify the smallest set of setting nodes that can explain the transition between the two beamline states in the learned embedding space.

3. Lacking explicit ground truth for node importance in accelerator data, the authors devise a principled statistical method to create a reliable proxy. They assess a node's importance by measuring its marginal contribution to reducing prediction error across all possible permutations of changes, providing a robust method for evaluation. The framework's effectiveness is validated on a large, real-world dataset from CEBAF operations.

**Audience:**

Yes

**Claims And Evidence:**

Yes

**Requested Changes:**

Please consider to include a brief discussion in the methodology or conclusion about the potential limitations of the permutation-based ground truth would add transparency. Acknowledging that this proxy averages out order-dependent effects and explaining why it is still a reasonable approach would strengthen the paper's integrity. In addition, the paper would be significantly more impactful if it included a short, concrete example of the framework in action.

**Strengths And Weaknesses:**

Pros.
1. The work tackles a complex, real-world problem where improvements can lead to significant savings in time and resources. Applying advanced explainability techniques to a physical system like a particle accelerator is both novel and highly practical.

2. The proposed three-model architecture is logical and well-designed. The separation of the prediction and explanation tasks is particularly clever, as it correctly models the physical constraint that operators can only adjust settings, while readings are outcomes. The use of an L1-regularized mask to encourage sparse, interpretable explanations is a well-suited choice for the problem.

3. The approach for establishing a proxy ground truth is a standout feature. This method addresses a critical challenge in XAI and allows for a quantitative and fair comparison against other methods. The experimental setup is thorough, including comparisons to multiple SOTA GNN explanation baselines, ablation studies, and parameter sensitivity analysis.

4. The proposed framework consistently outperforms baseline methods across multiple metrics, achieving the highest performance in Hits@5 (42.25%) and Precision@5 (33.13%). The ablation study further validates the design, confirming that both the graph-based structure and the embedding model are crucial for achieving optimal performance.

Cons:

The permutation-based method for generating ground truth is innovative but rests on the assumption that a node's contribution can be averaged across all possible ordering of changes. However, in a physical system, the order of adjustments can matter. The paper shows the "actual path" taken during an experiment (the blue trajectories in Fig. 4) but then creates the ground truth by averaging over all 720 possible paths. This might obscure order-dependent effects and should be discussed as a potential limitation of the proxy.

The explanation model's objective is to minimize the distance between two graph states in the learned embedding space. While technically sound, the paper does not fully motivate why this is a good proxy for physical similarity. A deeper justification for how the self-supervised learning objective (Eq. 4) ensures that the embedding space captures meaningful operational states would strengthen the argument of this paper.

---

> ### Author Response · Authors · 2025-09-02
> **Response to Reviewer B3WP**
>
> Dear Reviewer B3WP,
>
> Thank you for your thorough review and valuable suggestions for enhancing our work. We would like to addressyour concerns as follows:
>
> &nbsp;
>
>
>
>
>
> ### **Concern 1: Order-dependence in permutation-based ground truth**
>
> **Comment:** The permutation-based proxy averages over all possible orderings of node changes, which may obscure order-dependent effects.
>
> **Response:**
> We appreciate this insightful observation. We agree that the physical order of adjustments can influence outcomes in real accelerator operations. Ideally, we would evaluate every possible permutation and identify the most important node in each case. However, when dozens of nodes change, this is computationally infeasible. To address this, we adopt a two-stage strategy:
>
> 1. We first evaluate the effect of each setting node independently, ranking them by their marginal contribution to lowering the prediction error (MAE).
> 2. We then take the top-K nodes and exhaustively enumerate all permutations within this reduced set.
>
> In this way, order-dependence is explicitly considered among the most influential nodes, while still keeping the computation tractable. Rather than simply averaging over all permutations, our method applies a principled filter that balances computational feasibility with fairness in evaluating node importance. Importantly, this design also reflects the practical reality of accelerator operations, where operators often make iterative and non-standardized adjustments, making it difficult to define a strict sequential ground truth. We will revise Section 3.4 to more clearly describe this process and adjust the language around “average contributions” to avoid misunderstanding.
>
> ---
>
> &nbsp;
>
>
> ### **Concern 2: Embedding space justification**
>
> **Comment:** The objective minimizes distance between states in the learned embedding space, but why is this a good proxy for physical similarity?
>
> **Response:**
> Thank you for highlighting this. Our embedding optimization (Eq. 4) explicitly aligns distances in feature space with those in the latent embedding space, ensuring that operationally meaningful differences (e.g., in quadrupole strengths or cavity phases) are preserved. The embedding space therefore maintains a geometry consistent with the input features while also leveraging graph connectivity to capture causal dependencies. This alignment guarantees that minimizing latent distance corresponds to recovering a physically plausible state. We will expand Section 3.1 to more clearly justify this design choice.
>
> ---
>
> &nbsp;
>
>
> ### **Concern 3: Request for a concrete example of the framework in action**
>
> **Comment:** The paper would be significantly more impactful if it included a short, concrete example of the framework in action.
>
> **Response:**
> We appreciate your valuable suggestion. To provide an exemplar analysis, we refer to the case illustrated in Figure 4, where six setting nodes change between the two graphs. For this example, we exhaustively evaluated all 720 possible permutations of the six node changes and compared them with the results produced by our framework. Our framework ranked the nodes in the descending order of importance as 6, 1, 2, 4, 5, 3. Notably, both approaches consistently identified the same most influential node: the setting node associated with the last change, which shifted the value of PCA component 1 from 10 to −12. This agreement between the exhaustive permutation analysis and our framework highlights the validity of our explanation method. To improve clarity, we will revise the manuscript to explicitly describe this example in the main text, enabling readers to better connect the framework with a real operational scenario.
>
> ---
>
>
> &nbsp;
>
> We sincerely hope our response could address your concerns. Looking forward to your reply!
>
> Best,
> Authors

---

> > ### Comment · Reviewer_B3WP · 2025-09-28
> > **Thanks for your response.**
> >
> > Thanks for your response! I found the responses have already addressed my concerns.

---

### Author Response · Authors · 2026-01-04
**TMLR Submission 4836**

Dear Action Editor Dr. Archambeau and the TMLR Editorial Team,

Our manuscript (submission 4836) has been under review since late May. We promptly responded to reviewer feedback in early September. After 12 weeks we had not received any updates and asked for a status update (Nov. 24). We received a response that a "decision will follow shortly". It has now been an additional 6 weeks without any further update.

We would be grateful to get a status update and an editorial decision soon.

We appreciate your time and your support of the TMLR review process.

Best,
The Authors

---

> ### Comment · Action_Editor_hU2R · 2026-01-05
> **Re: TMLR Submission 4836**
>
> Dear Authors,
>
> Apologies for the delay. This is due to one of the reviewers being unresponsive. I'll post my decision shortly.
>
> Best wishes,
>
> AE

---

### Comment · Action_Editor_hU2R · 2026-01-29
**Final version check**

Dear authors,

Can you please indicate the changes you made to final version of the paper? A few outstanding issues need to be addressed (see decision). Once this is clarified I will be able to approve the camera ready version.

Thank you,

AE

---

> ### Author Response · Authors · 2026-02-02
>
> Dear Action Editor,
>
> We have revised the paper according to your comment in "Decision by Action Editor hU2R", particularly Concern 1,3,5 of Reviewer L9ir and all concerns by Reviewer B3WP and GpC3.
>
> We believe that all the issues mentioned in the decision have been solved.
>
> Thank you so much for your time and effort in reviewing our paper.
>
> Best,
> The Authors

---

### Decision · Action_Editor_hU2R · 2026-01-05

**Recommendation:** Accept with minor revision

**Additional Comments:**

The authors promised a series of updates and clarifications in their response. Hence, I kindly request that the authors submit a revised version of their paper in which the following issues are addressed and highlighted:

1/ A clarification of an outstanding concern from reviewer L9ir raised during the discussion phase: "The explainer and the ground-truth ranking are derived from the same learned function, introducing a shared inductive bias. This violates a core principle that the evaluation metric should not be derived from the model itself."
2/ Concerns 1-3 raised by reviewer B3WP.
3/ Critical and non-critical comments raised by reviewer GpC3.
4/ Concern 1 and 5 raised by reviewer L9ir.

**Audience:**

Yes

**Audience Explanation:**

This paper tackles an interesting and challenging application of GNNs. The proposed solutions are specific to the problem at hand, but could be generalised to other settings as the authors explain in their response. Hence, the results reported in this work are of relatively broad interest to the community. The authors further propose a methodology for the reliable evaluation of the method as ground truth is not readily available.

**Claims And Evidence:**

Yes

**Claims Explanation:**

This work tackles the challenging task of beam tuning at the Continuous Electron Beam Accelerator Facility (CEBAF). The work develops an explanation-driven framework leveraging GNNs to solve this problem. In particular, it helps identifying which setting adjustments are most responsible for a change between two different beamline states.

Reviewers overall appreciated this work, highlighting the following contributions:
1/ The work tackles a complex, real-world problem, applying advanced explainability techniques to a physical system like a particle accelerator.
2/ The proposed is sound and the modelling approach of separating the prediction and explanation tasks is innovative.
3/ Extensive comparison demonstrating gains compared to competing GNN solutions.

The main concerns resided on the validity of the ground truth approximation and the evaluation methodology. Overall, the authors addressed the concerns raised by the reviewers, promising a number of changes and additions. However, they did not submit a revised version with these changes (see Additional Comments).